# Occurrence of virulence genes in multidrug-resistant *Escherichia coli* isolates from humans, animals, and the environment: One health perspective

Edwin M. James[1,2], Zuhura I. Kimera[3], Fauster X. Mgaya[4], Elieshiupendo M. Niccodem[ORCID][4,5]*, Joely Ezekiely Efraim[6], Mecky I. Matee[4], Erasto V. Mbugi[ORCID][1]

1 Department of Biochemistry and Molecular Biology, Muhimbili University of Health and Allied Sciences, Dar es Salaam, Tanzania, 2 Department of Biochemistry and Molecular Biology, Kilimanjaro Christian Medical University College, Kilimanjaro, Tanzania, 3 Department of Environmental and Occupational Health, Muhimbili University of Health and Allied Sciences, Dar es Salaam, Tanzania, 4 Department of Microbiology & Immunology, Muhimbili University of Health and Allied Sciences, Dar es Salaam, Tanzania, 5 Department of Microbiology & Immunology, Kilimanjaro Christian Medical University College, Kilimanjaro, Tanzania, 6 Department of Economics and Statistics, Moshi Cooperative University, Kilimanjaro, Tanzania

* eshimankamike137@gmail.com

**Data Availability Statement:** All relevant data are within the manuscript and its Supporting Information files.

## Abstract

*Escherichia coli* is one of the critical One Health pathogens due to its vast array of virulence and antimicrobial resistance genes. This study used multiplex PCR to determine the occurrence of virulence genes *bfp*, *ompA*, *traT*, *eaeA*, and *stx1* among 50 multidrug-resistant (MDR) *E. coli* isolates from humans (n = 15), animals (n = 29), and the environment (n = 6) in Dar es Salaam, Tanzania. Their association with antimicrobial-resistant genes (ARGs) was determined using Principal Component Analysis (PCA). All 50/50 (100%) MDR *E. coli* isolates carried at least one virulence gene, with 19/50 (38%) carrying four genes, *bfp* + *traT* + *eaeA* + *ompA*. The findings showed a high occurrence of virulence genes *bfp* (82%), *traT* (82%), *eaeA* (78%), and *ompA* (72%); the study detected no *stx1* in any of the isolates. In humans, the most detected virulence genes were *bfp* and *traT* 14/15 (93.3%); for poultry, it was *eaeA* 13/14 (92.9%); for pigs, was *bfp* and *traT* 13/15 (86.7%); while for river water, it was *eaeA* 6/6 (100%). The study observed no significant association between virulence genes and ARGs. PCA results show the genes *ompA*, *traT*, *eaeA*, and *bfp* contributed to the virulence of the isolates, and *blaTEM*, *blaCTX-M*, and *qnrs* contributed to ARGs. The PCA ellipses show that isolates from pigs had more virulence genes than those isolated from poultry, river water, and humans. The high frequency of numerous virulence genes in MDR *E. coli* isolates from humans, animals, and the environment indicates that these isolates have a very high potential to cause diseases that are difficult to treat because they are MDR.

## Background

Gram-negative bacterial pathogens, including third-generation cephalosporin-resistant *Escherichia coli* (*E. coli*) cause severe infections due to their high virulence and antimicrobial

**Funding:** The author(s) received no specific funding for this work.

**Competing interests:** The authors have declared that no competing interests exist.

resistance, with limited treatment options [1,2]. This group of bacteria possesses a wide array of antimicrobial resistance genes (ARGs) and virulence genes (VGs), making them particularly dangerous pathogens [3].

*E. coli* carries several antimicrobial resistance genes (ARGs) that include; extended- spectrum beta-lactamase (ESBL) genes (*blaSHV*, *blaCTX-M* and *blaTEM*), carbapenemase genes (*blaKPC*, *blaNDM*, and *blaOXA-48*), plasmid-mediated quinolone resistance (PMQR) genes (*qnrA*, *qnrB*, *qnrS*, *qnrC*, *qnrD*, *qepA*, and *aac(6)-Ib-cr*), and tetracycline resistance genes (*tetA*, *tetB*) [4,5]. Other genes encode for resistance against macrolides (*ermB*), sulfonamides (*sul1*, *sul2*, *sul3*), trimethoprim (*dfrA*), aminoglycosides (*aac(3)*, *aph(3)*, *aadA*) and chloramphenicol (*catA1*) [4–6].

In addition, *E. coli* has several virulence mechanisms that include adherence to the host cell, membrane, invasion into the host cell, competition for iron, toxin production, and host immune evasion, among others [5–8]. The bundle-forming pilus (*bfp*) gene and *E. coli* attaching and effacing (*eae*) gene are involved in adherence to a host cell, *stx-1* encodes production of the Shiga toxin (*stx*), while the outer membrane protein A (*ompA*) gene and the *traT* gene help *E. coli* to evade host immune system [5–10].

Both antimicrobial resistance and virulence genes are essential for pathogenic bacteria to adapt to and survive in competitive microbial environments [11,12], and their co-occurrence leads to increased severity of infections [12]. Several studies have reported the occurrence and distribution of ARGs, virulence genes, or both of them in multidrug-resistant (MDR) *E. coli* isolates from humans, animals, and the environment [4,6,13–17]. Fewer studies have gone further to assess the association between ARGs and virulence genes showing significant variations [5,13,18–23].

In Tanzania, there have been reports of extensive interaction among humans, animals, and the environment, which has the potential to favor horizontal transmission of ARGs and virulence genesacross these compartments [4,24–30]. However, only one has assessed the occurrence and distribution of virulence genes in MDR *E. coli* and their association with ARGs in one health compartment [5]. In that study, *E. coli* isolated from humans, rodents, chickens, and soil had predominantly *blaTEM*, *blaCTX-M*, *blaSHV*, *tetA*, *tetB*, and *qnrA* ARGs and virulence genes (*traT*) and did find positive correlations with virulence genes *qnrA*, *qnrB*, and *bfp* [5]. The study however, showed that rodent isolates had more antimicrobial and virulence genes than those isolated from chickens, soil, and humans, underlying a rather complex epidemiology of resistance and virulence determinants.

This study was carried out to screen for the occurrence of virulence genes *bfp*, *ompA*, *traT*, *eaeA*, and *stx1* in MDR *E. coli* isolates from humans, animals, and the environment and to determine their association with ESBL genes (*blaCTX-M*, *blaTEM*, and *blaSHV*) and plasmid-mediated quinolone resistance (PMQR) genes (*qnrA*, *qnrB*, *qnrS*, *qnrC*, *qnrD*, *qepA*, and *aac (6)-Ib-cr*). To demonstrate the influence of resistance genes on the occurrence of different virulence genes, principal component analysis (PCA) was used, and with the deployment of PCA ellipses, it was possible to find whether isolates from humans, pigs, poultry, or river water possessed more virulence genes or not. This study was able to assess the respective capability of these compartments as gene reservoirs. The study hypothesized a high frequency of virulence genes in these isolates since virulence genes and ARGs tend to co-occur.

## Materials and methods

The study attained ethical clearance from the Research and Ethical Committee of the Muhimbili University of Health and Allied Sciences (MUHAS) (Ref. No. MUHAS-REC-03-2024-2116).

## Study location

The study was conducted in Dar es Salaam, Tanzania. The city has five administrative districts, with a population of 5,383,728 living in 1,393 km$^2$, a population density of 3,865/km$^2$, and an annual population growth of 2.1%. It is the main engine of economic development (2022 population census).

## Study isolates

This cross-sectional study involved 50 isolates of MDR *E. coli* that were collected in previous studies that were conducted in Dar es Salaam [26,29,31]. Fifteen isolates were from humans, 14 from poultry, 15 from domestic pigs (15), and 6 from the river (6). The isolates had been stored at -80˚C at the Microbiology Research Laboratory, of the Muhimbili University of Health and Allied Sciences (MUHAS).

## Biochemical identification of the isolates

The isolates were identified by colonial morphology, Gram stain, and a set of conventional biochemical tests (catalase, oxidase, indole, methyl red, Voges–Proskauer and citrate utilization tests, and lactose fermentation) [4,26,29,31]. Confirmation was done using the Analytical Profile Index (API) 20E (BioMérieux, Marcyl'Etoile, France).

## Phenotypic and genotypic antimicrobial susceptibility testing

Phenotypic antimicrobial susceptibility testing was done using the Kirby–Bauer disc diffusion method [32] on Mueller Hinton agar (Becton, Dickinson and Company, New Jersey, USA) based on the 2022 Clinical Laboratory Standard Institute (CLSI) guidelines [33] and using the most commonly prescribed antibiotics. An isolate was considered multidrug-resistant (MDR) if it was non-susceptible to three or more different classes of antimicrobials [34].

Genotypic screening of ARGs was done as described in a previous study [4]. Briefly, DNA was extracted from cultured isolates using the boiling method. The polymerase chain reaction (PCR) was used for screening ESBL genes (*blaCTX-M*, *blaTEM*, and *blaSHV*) and plasmid-mediated quinolone resistance (PMQR) genes (*qnrA*, *qnrB*, *qnrS*, *qnrC*, *qnrD*, *qepA*, and *aac (6)-Ib-cr*) [4].

## DNA extraction for the screening of virulence genes

In screening for virulence genes, MDR *E. coli* isolates were inoculated on MacConkey agar (MCA) (Oxoid Ltd., Hampshire, UK) and incubated aerobically at 37˚C for 24 hrs. Colonies that appeared pink or red on MCA were subcultured on nutrient agar (NA) (HI Media, Mumbai, India) and incubated aerobically at 37˚C for 24 hours [12]. DNA extraction was done by boiling in a water bath at 100˚C for 10 min and centrifugation at 13000 rpm for 10 min [35]. Centrifugation and separation of the supernatant were done in sterile Eppendorf PCR tubes (Eppendorf AG, Hamburg, Germany). The concentration and quality of DNA were measured using a Nanodrop spectrophotometer (Biochrom LTD, Cambridge, England) at a 260/280 wavelength. The researcher stored the extracted DNA at -20˚C before using it [4].

## PCR mixture for the detection of virulence genes

For the detection of virulence genes *bfp*, *traT*, *ompA*, *stx1, and eaeA*, this study used the One Tag Master Mix Hot Start DNA polymerase kit (New England Biolabs, Ipswich, MA, USA). Each gene's forward and reverse primers are shown in Table 1. Lyophilized primers for target

**Table 1. Primers for virulence genes *bfp*, *traT*, *eaeA*, *ompA*, and *stx1*.**

| Target gene | Primers | Primer Sequences (5'—-3') | Annealing temp (˚C) | Amplicon size (bp) | References |
|---|---|---|---|---|---|
| *ompA* | *ompA-F* | AGCTATCGCGATTGCAGTG | 52 | 919 | [5] |
|  | *ompA-R* | GGTGTTGCCAGTAACCGG |  |  |  |
| *stx1* | *stx1-F* | TCTCAGTGGGCGTTCTTATG | 58 | 388 | [5] |
|  | *stx1-R* | TACCCCCTCAACTGCTAATA |  |  |  |
| *Bfp* | *bfp-F* | AATGGTGCTTGCGCTTGCTGC | 56 | 324 | [5] |
|  | *bfp-R* | GCCGCTTTATCCAACCTGGTA |  |  |  |
| *traT* | *traT-F* | GGTGTGGTGCGATGAGCACAG | 63 | 290 | [5] |
|  | *traT-R* | CACGGTTCAGCCATCCCTGAG |  |  |  |
| *eaeA* | *eaeA-F* | ATGCTTAGTGCTGGTTTAGG | 58 | 248 | [5] |
|  | *eaeA-R* | GCCTTCATCATTTCGCTTTC |  |  |  |

genes were reconstituted using nuclease-free water to obtain 100 μM stock and 10 μM working solutions before storage at -20˚C.

## PCR conditions for the detection of *ompA*, *traT*, *stx1*, *eaeA*, and *bfp* genes

PCR was carried out in a total volume of 25 μL containing 12.5 μL of 2X Taq PCR Master Mix, 0.5 μL of the 10 μM forward primer for each gene, and 0.5 μL of the 10 μM reverse primer for each gene, 2 μL of DNA template, and 8.5 μL nuclease-free water. PCR optimization and conditioning are indicated in Table 2.

## Visualization of PCR products by electrophoresis

For visualization of PCR products, 1X Tris-acetate–EDTA (TAE) working buffer was prepared by mixing 980 mL of distilled water with 20 mL of 50X TAE buffer stock solutions in a conical flask. Then, 1.5% of the agarose gel was prepared by dissolving 1.5 grams of agarose (Merck, SA) in 100 mL of the working 1X TAE buffer solution, and the mixture was heated in a microwave until completely dissolved to form homogenous [4]. To facilitate visualization of DNA, 5 μLgel red stain (Sigma-Aldrich, USA) was added to the agarose gel, and cooled to about 60˚C, before pouring into the sample comb's casting tray and allowed to solidify at room temperature [4]. For DNA band size estimation, the amplicons of each sample were mixed with 2 μL loading dye and a 100-base pair marker (New England Biolabs, Ipswich, MA, USA). All gels were run in 1X TAE buffer at 120V for 60 minutes and visualized by UV trans-illumination [4].

## Data management and analysis

Data was entered and analyzed in a data spreadsheet (Microsoft® Office Excel 2016). Descriptive analysis was used to determine the frequency of virulence gene(s) among MDR *E. coli*.

**Table 2. Multiplex PCR conditions for amplification of virulence genes in MDR *E. coli* isolates.**

| Program | Targeted Genes | Amplification conditions for virulence genes | | | | | |
|---|---|---|---|---|---|---|---|
|  |  | Initial Denaturation | Denaturation | Annealing | Primer extension | Final extension | Number of cycles |
| PCR 1 | *ompA*, *traT* | 94˚C for 5min | 94˚C for 1min | 58˚C for 30sec | 68˚C for 3min | 72˚C for 10min | x30 |
| PCR 2 | *stx1*, *eae* | 95˚C for 5min | 95˚C for 30sec | 55˚C for 30sec | 72˚C for 3min | 72˚C for 7min | x35 |
| PCR 3 | *Bfp* | 95˚C for 5min | 95˚C for 30sec | 55˚C for 30sec | 72˚C for 3min | 72˚C for 7min | x35 |

**Table 3. Distribution of virulence genes in MDR *E. coli* isolates from different sample sources.**

| Genes | Different Sample Sources n (%) | | | | Total (n = 50) | p-value |
|---|---|---|---|---|---|---|
| | Human (n = 15) | Poultry (n = 14) | Pigs (n = 15) | River water (n = 6) | | |
| *ompA* | 12 (80%) | 10 (71.4%) | 10 (66.7%) | 4 (66.7%) | 36 (72%) | 0.855 |
| *traT* | 14 (93.3%) | 11 (78.6%) | 13 (86.7%) | 3 (50%) | 41 (82%) | 0.122 |
| *eaeA* | 13 (86.7%) | 13 (92.9%) | 7 (46.7%) | 6 (100%) | 39 (78%) | 0.005 |
| *Bfp* | 14 (93.3%) | 9 (64.3%) | 13 (86.7%) | 5 (83.3%) | 41 (82%) | 0.062 |

The differences in the occurrence of the five genes between sample sources were determined using Chi-square, and a p-value of $< 0.05$ was considered significant. The Spearman rank correlation test was used to elucidate the association between virulence genes and ARGs. Principal component analysis (PCA) was performed using the R statistical software version 4.1.3 (2022-03-10) to assess the relationships between antimicrobial resistance and virulence genes in the different sample sources.

## Results

### Distribution of virulence genes in MDR *E. coli* isolates from different sample sources

Overall, all MDR *E. coli* isolates, 50/50 (100%), carried at least one of the virulence genes, with 19/50 (38%) carrying four types of virulence genes, *eaeA* + *bfp* + *ompA* + *traT*. The distribution of virulence genes was as follows: *bfp* (82%), *traT* (82%), *eaeA* (78%), and *ompA* (72%), while *stx1* was not detected (Table 3). For humans, the most detected virulence genes were *bfp* and *traT* 14/15 (93.3%); for poultry, it was *eaeA* 13/14 (92.9%); for pigs, it was *bfp* and *traT* 13/15 (86.7%); while for river water the most detected gene was *eaeA* 6/6 (100%) (Fig 1). The

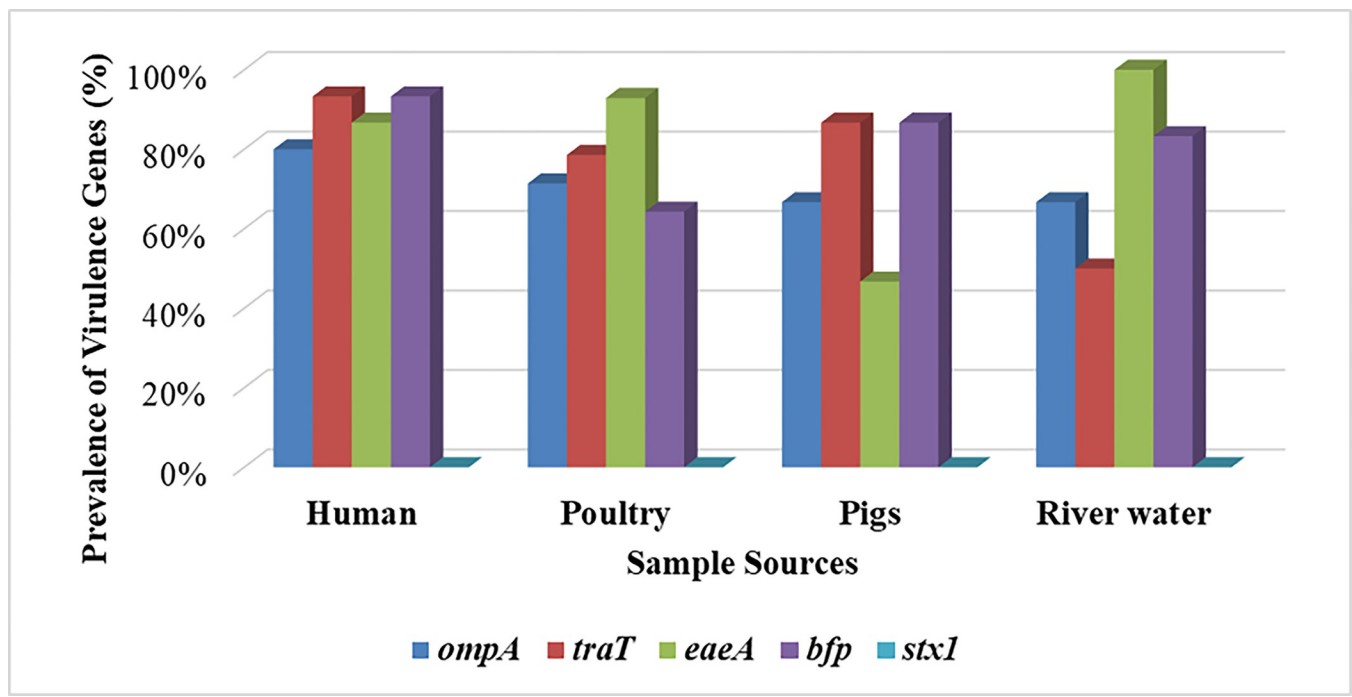

**Fig 1. Frequencies of virulence genes in MDR *E. coli* isolates from different sample sources.**

**Table 4. Co-occurrence of virulence genes in MDR *E. coli* isolates from different sample sources.**

| Virulence Genes | Different Sample Sources n (%) | | | | Total n = 50 (%) |
|---|---|---|---|---|---|
| | Humans n = 15(%) | Poultry n = 14(%) | Pig n = 15 (%) | River Water n = 6 (%) | |
| *ompA* | 0 (0) | 0 (0) | 1 (6.7) | 0 (0) | 1 (2) |
| *traT* | 0 (0) | 0 (0) | 1 (6.7) | 0 (0) | 1 (2) |
| *ompA+bfp* | 0 (0) | 0 (0) | 1 (6.7) | 0 (0) | 1 (2) |
| *traT+bfp* | 2 (13.3) | 0 (0) | 1 (6.7) | 0 (0) | 3 (6) |
| *traT+eaeA* | 0 (0) | 1 (7) | 2 (13.3) | 0 (0) | 3 (6) |
| *ompA+eaeA* | 0 (0) | 0 (0) | 0 (0) | 1 (16.7) | 1 (2) |
| *bfp+traT+eaeA* | 1 (6.7) | 3 (21) | 2 (13.3) | 2 (33.3) | 8 (16) |
| *ompA+bfp+eaeA* | 1 (6.7) | 4 (29) | 0 (0) | 2 (33.3) | 7 (14) |
| *ompA+traT+bfp* | 0 (0) | 0 (0) | 4 (26.6) | 0 (0) | 4 (8) |
| *ompA+traT+eaeA* | 1 (6.7) | 0 (0) | 1 (6.7) | 0 (0) | 2 (4) |
| *ompA+traT+bfp+eaeA* | 10 (66.6) | 6 (43) | 2 (13.3) | 1 (16.7) | 19 (38) |

difference in the distribution of the virulence gene *eaeA* in different sample sources was statistically significant (p = 0.005), while the distribution of other virulence genes had no statistical significance.

## Co-occurrence of virulence genes in MDR *E. coli* isolates from different sample sources

As shown in Table 4 below, 38% of the isolates were found to carry four virulence genes, *ompA + traT + bfp + eaeA*. The majority of the isolates (42%) were found to carry three different types of virulence genes, *bfp+traT+eaeA* (16%) and *ompA+bfp+eaeA* (14%) being the most frequent combination of three genes observed. The co-occurrence of two genes was observed in 16% of the isolates, where *traT+eaeA* (6%) and *traT+bfp* (6%) were most frequently observed. Only two (4%) isolates were found to carry only one type of virulence gene, *ompA* (2%) and *traT* (2%), and both isolates were from pigs.

## Association between virulence genes and ARGs

As indicated in Table 5, no significant associations existed between virulence genes and ARGs. The associations were either weakly positive or weakly negative.

Figs 2–4 show the gel electrophoretic bands of the virulence genes *ompA* and *traT*, *eaeA* and *stx1*, and *bfp*, respectively. The same figures show positive and negative samples for the genes and negative and positive controls.

**Table 5. Association between resistance and virulence genes of MDR *E. coli* isolates.**

| AMR Genes | Virulence Genes | | | |
|---|---|---|---|---|
| | Correlation Coefficients (r) | | | |
| | *ompA* | *traT* | *eaeA* | *Bfp* |
| *bla-CTX-M* | -0.08 | -0.21 | 0.23 | -0.3 |
| *bla-TEM* | 0.39 | -0.21 | 0.23 | 0.27 |
| *qnrB* | -0.11 | 0.18 | -0.33 | -0.26 |
| *qnrS* | -0.35 | 0.25 | 0.24 | 0.12 |

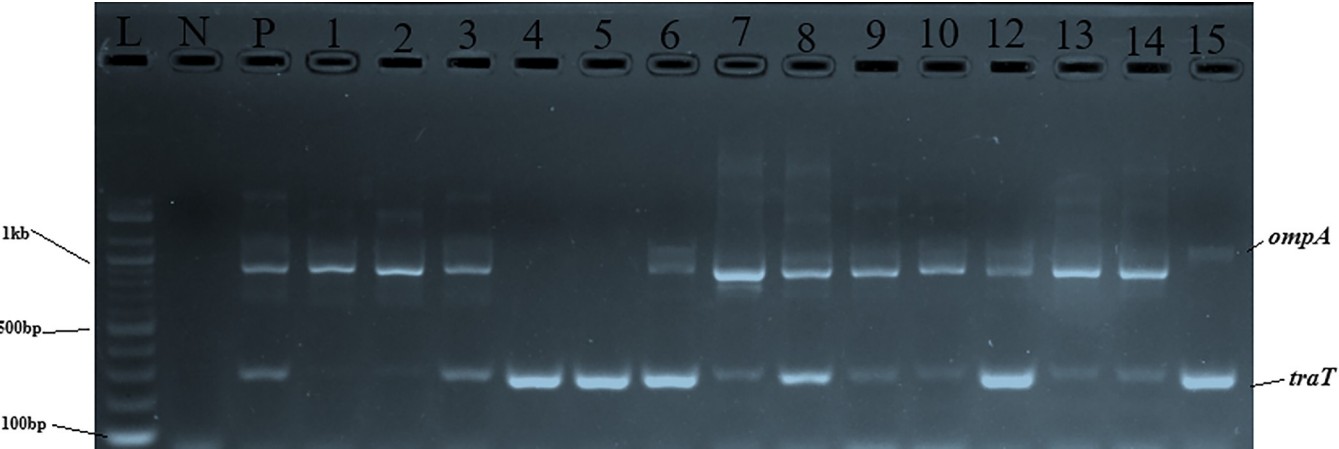

**Fig 2. Gel electrophoretic bands of virulence genes (*ompA* and *traT*).** Letters L–DNA ladder, N–negative control, and P–positive control. Numbers 3,6,7,8,9,10,12,13,14 and 15 are positive samples for both genes. Numbers 1 and 2 are positive samples for the *ompA* gene while 4 and 5 are positive samples for the *traT* gene.

## Principal component analysis results

As shown in Fig 5, the arrows (vectors) for the *ompA* and *traT* genes align closer to PC1, indicating a more significant and positive correlation between the virulent genes and PC1, while vectors for the *ompA* and *traT* genes are pointing in opposite directions, indicating a negative correlation between them. The length of the arrows shows that the *ompA* gene contributed

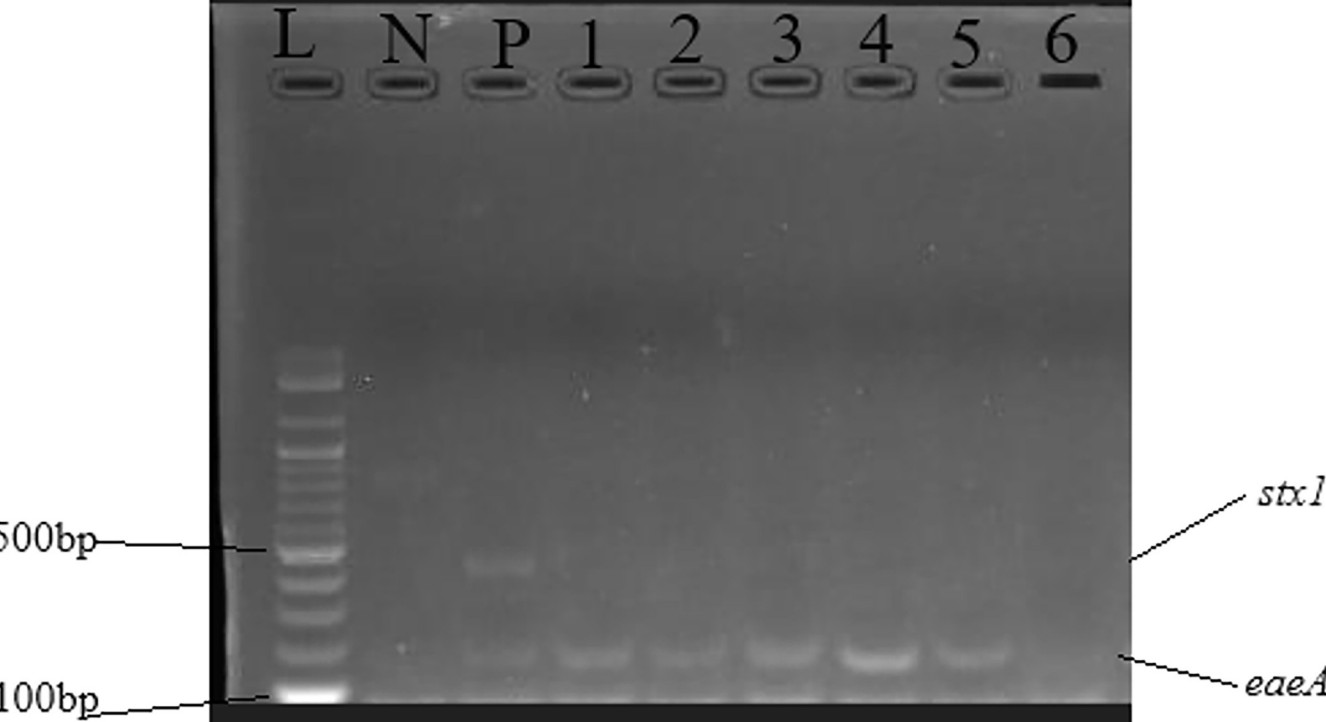

**Fig 3. Gel electrophoretic bands of virulence genes *eaeA* and *stx1*.** Letters L–DNA ladder, N–negative control, and P–positive control. Numbers 1,2,3,4 and 5 are positive samples for *eaeA* gene. Number 6 is the negative sample for both genes.

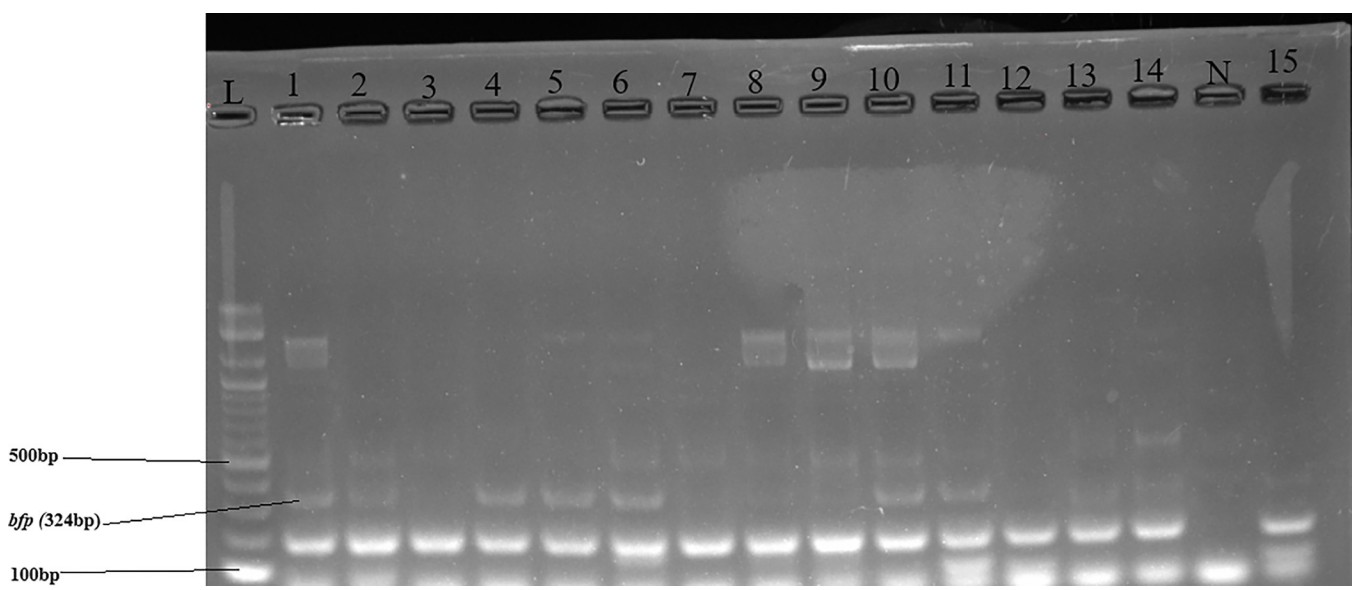

**Fig 4. Gel electrophoretic bands of virulence gene *bfp*.** Letters L–DNA ladder, N–negative control. Numbers 1,2,4,5,10,11,13 and 14 are positive samples. Numbers 3,7,8,9,12 and 15 are negative samples.

**Fig 5. Principal component analysis for virulence genes of *E. coli* isolates.**

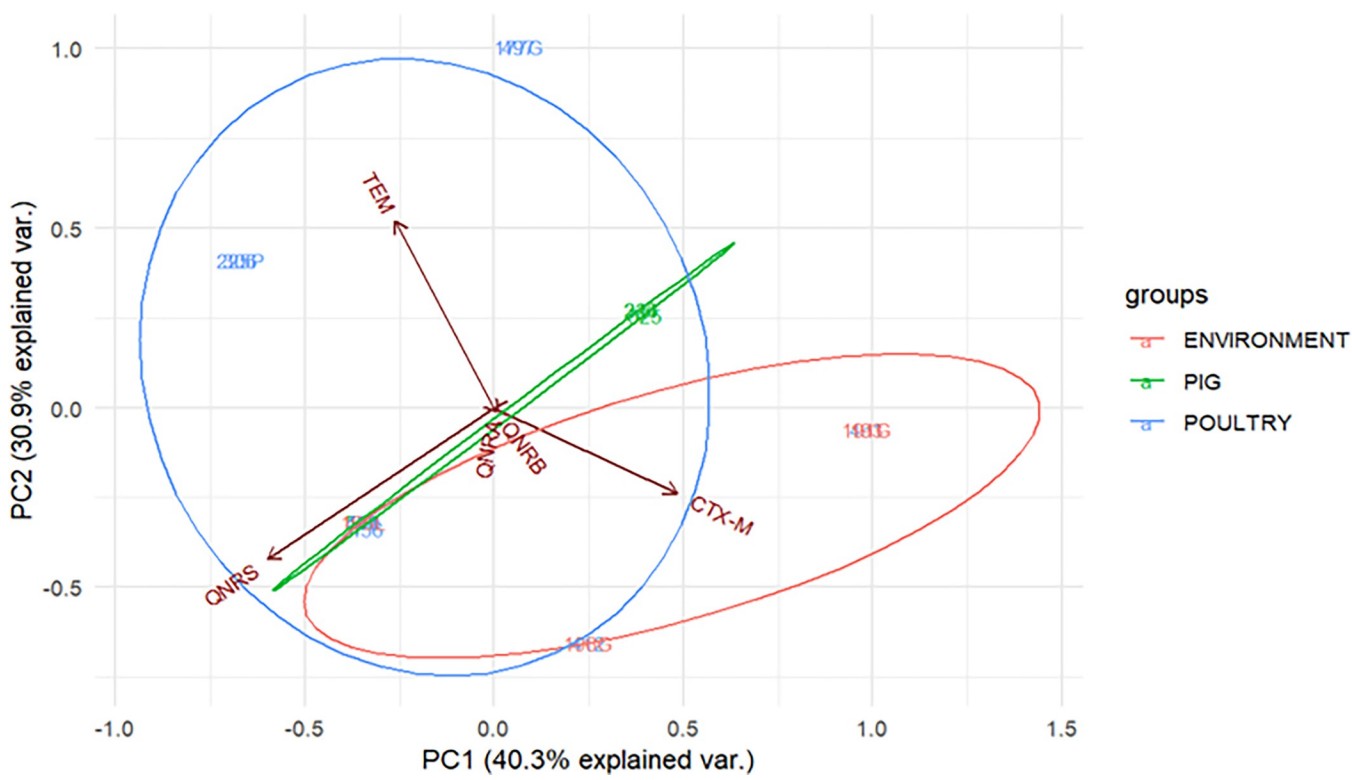

**Fig 6. Principal component analysis of resistance genes of MDR *E. coli* isolates.**

more to the virulence of isolates than *traT*. The vectors for the *bfp* and *eaeA* genes are close to each other and PC2, showing their influence on virulence. The length of vectors indicates that the same gene had a more substantial impact on the virulence of isolates than the *bfp* gene.

The vectors for *blaTEM*, *blaCTX-M*, and *qnrS* genes show that these genes contributed more to the resistance of isolates than other ARGs (Fig 6). The ellipses' sizes indicate that poultry isolates had the highest prevalence of ARGs, followed by environmental isolates, with pig isolates having the lowest prevalence.

PC1 indicates genes with the most significant difference in occurrence across sample sources. PC2 indicates the other significant difference not covered by PC1. Arrows indicate the original variable (the virulence genes of the isolates), and ellipses indicate a region that contains 95% of all samples of a particular source.

PC1 indicates genes with the most significant occurrence difference across sample sources. PC2 indicates the other considerable difference not covered by PC1. Arrows indicate the original variable (ARGs of the isolates), and ellipses indicate a region that contains 95% of all samples of a particular source.

## Discussion

This study determined the occurrence and distribution of virulence genes *bfp*, *traT*, *ompA*, *stx1*, and *eaeA*, as well as their association with ESBL genes (*blaCTX-M*, *blaTEM*, and *blaSHV*) and plasmid-mediated quinolone resistance (PMQR) genes (*qnrA*, *qnrB*, *qnrS*, *qnrC*, *qnrD*, *qepA*, and *aac (6)-Ib-cr*) in MDR *E. coli* isolates from humans, animals, and the environment in Dar es Salaam, Tanzania. The influence of resistance genes on the occurrence of different virulence genes was demonstrated using principal component analysis (PCA). To find out if

isolates from humans, pigs, poultry, or river water had more virulence genes, PCA ellipses were employed. It is evident from this study that the respective compartments have potential capabilities to serve as gene reservoirs.

The study found that all 50/50 (100%) of the MDR *E. coli* isolates carried at least one virulence gene, with 19/50 (38%) having four genes. This study's results align with a study in Karatu, Tanzania, showing a high proportion of virulence genes (72%) in the MDR *E. coli* isolates from humans, animals, and the environment [5]. Comparatively, this study found a high frequency of virulence genes *bfp* (82%), *traT* (82%), *eaeA* (78%), and *ompA* (72%). The study in Karatu had a similar distribution of virulence gene *ompA*, 72%, but lower occurrences of virulence genes *traT* (26%), *bfp* (10%), and *eaeA* (2%) [5]. In both studies, the occurrence of virulence gene *stx1* was low, being 4% in Karatu and 0% in this study, which is in keeping with those of studies conducted in Egypt [36] and Turkey [37], which did not find the gene in *E. coli* isolates from meat workers and chicken carcasses, respectively. According to the risk factors linked to the *stx*1 and *stx*2 genes of *E. coli*, these virulence genes, which are spread by highly mobile bacteriophages, may have different ecologies, and their frequency is influenced by seasonal and environmental factors, which explains the variations observed globally [38].

The most frequently detected virulence genes in the isolates from humans were *traT* (which encodes for outer membrane protein [39]) and *bfp*, which confer the ability to adhere to host cells using bundle-forming pili [5]. Additionally, 86.7% of *E. coli* isolates from human samples harbored the *eaeA* gene, which encodes intimin involved in *E. coli* adherence to the host cell [40]. The *ompA* gene, encoding for outer membrane protein A, helps *E. coli* to evade the host immune system [41] and was detected in 80% of isolates. These four genes, which occurred in 80% to 93% of the isolates from humans, seem to have a high significance in human infections in this area.

Compared with other studies, the prevalence of *traT* seen in this study is higher than that reported in studies done in Karatu (28.6%), Slovakia (59.4%), and Lithuania (81.3%) [5,20,42]. Likewise, the frequency of the virulence gene *bfp* (93.3%) found by this study in the human isolates is higher than that reported in Karatu (0%) and Syria (42%) [5,43].The *eaeA* gene was found in a higher proportion of isolates than the 42% reported in Syria [5,43], but nearly equal to the 71.4% reported in another study conducted in Tanzania [5].

In poultry isolates, the same genes were the most frequently detected, with *eaeA* present in 92.9% of the samples, followed by *traT* (78.9%), *ompA* (71.4%), and *bfp* (64.3%), which differ from a study in Karatu in which the prevalence of virulence genes was *eaeA* (0%), *traT* (33.3%), *ompA* (100%), *bfp* (0%) and *stx1* (8.3%) [5], and in China, where 60% of poultry isolates carried the *traT* gene [44], and in Iraq, 22% of poultry isolates harbored the *bfp* gene [45]. The high prevalence of *eaeA* found in poultry, a gene that encodes for the intimin protein, suggests a significant risk of *enteropathogenic E. coli* (EPEC) infections from poultry [46].

In isolates from pigs, this study found frequencies of virulence genes *traT* (86.7%), *bfp* (86.7%), *ompA* (66.7%), and *eaeA* (46.7%), a higher frequency of the *eaeA* gene compared to that found in rabbits (28.3%) in Tunisia [47]. However, these findings are similar to studies conducted in China, which reported that *ompA* (86.9%) and *traT* (84.9%) were among the most frequently detected virulence genes in *E. coli* isolates from animals [48]. In Pakistan, the prevalence of the *eaeA* gene (90%) and *stx1* gene (25%) in animals [49] was higher than the ones reported in this study. The high presence of *traT*, *bfp*, *ompA*, and *eaeA* genes in pig isolates means they can adhere to host cells and evade the host immune system, contributing to their persistence and pathogenicity.

This study detected the *eaeA* gene in all river samples (100%), underscoring the environmental persistence of EPEC strains. The distribution of the other virulence genes was *bfp* (83.3%), *ompA* (66.7%), *traT* (50%), and *stx1* (0%). In Pakistan, the *eaeA* gene was found in

(100%) of *E. coli* isolates from the environment [49]. In comparison, the prevalence was low (16.8%) in Thailand of *eaeA* in seawater *E. coli* isolates [40]. Collectively, results from this study and those of others regions strongly indicate the virulence potential of *E. coli* in water sources and the consequences for public health.

This study found a significant difference in the distribution of the *eaeA* gene across different sample sources (p = 0.005), suggesting specific varied epidemiological patterns and transmission dynamics in the sampled environments [18,47,49,50]. In contrast, this study found no significant difference in the distribution of other virulence genes, *bfp*, *traT*, and *ompA*, across different sources. This uniformity might indicate a broader adaptive advantage conferred by these genes, enabling *E. coli* to thrive in diverse environments [12,42,48].

Using principal component analysis, this study found the distribution of the virulence gene *ompA* to be most important in differentiating samples from different sources, followed by the virulence genes *traT*, *eaeA*, and *bfp*. The vectors for virulence genes *ompA* and *traT* were close to PC1 (which indicates the size of the difference among the different sample sources), and those for *eaeA* and *bfp* were close to PC2 (which shows the size of the difference among the different sample sources that PC1 did not capture). Using PC1 and PC2 together, virulence genes *ompA*, *traT*, *eaeA*, and *bfp* were shown to contribute to the virulence of these isolates, aligning with the findings of another study that found the vectors of virulence genes *traT* and *bfp* were near PC1 and PC2, respectively [5].

Based on the different sizes of ellipses, PCA showed that isolates from pigs had a higher proportion of virulence genes, followed by those isolated from the environment, with those from poultry and humans having the lowest proportion, which indicates their respective potential as a reservoir for the genes [5,51].

Using the Spearman rank correlation test, this study found either weak positive or negative non-significant associations between virulence genes and ARGs. The association between *ompA* and *blaTEM* was weakly positive (r = 0.39), while that between *ompA* and *qnrS* was weakly negative (r = -0.35). The association between other virulence genes and ARGs ranged between r = 0.39 and -0.35. This observation may be due to the complex independent evolution and horizontal transfer of virulence genes and ARGs [11]. Indeed, some studies have suggested that the acquisition of resistance to certain antibiotics may be associated with an increase or decrease in the virulence levels depending on the location and mechanism of transfer of specific genes [5,18,52–54].

In summary, this study provides valuable insight into the occurrence of selected virulence and antimicrobial resistance (AMR) genes in MDR *E. coli* isolates from humans and some isolates from poultry, pigs, and river water. This study acknowledges that isolates missed information on ARGs. This study also used a limited set of virulence and AMR genes, which may have contributed to the observed lack of association between virulence genes and ARGs. Finally, this study only used PCR, not advanced genomics such as whole genome sequencing (WGS) and metagenomics. This would have helped in understanding the complexity of resistors' flow across the compartments and informed interventional measures.

## Conclusion

This study demonstrates the widespread distribution of virulence genes *bfp*, *traT*, *eaeA*, and *ompA* in MDR *E. coli* isolates from humans, animals, and the environment. These genes encode for various mechanisms of virulence. This study found significant differences in the distribution of the *eaeA* gene across different sample sources, suggesting specific varied epidemiological patterns and transmission dynamics in the sampled environments. In contrast, this study found no significant difference in the distribution of other virulence genes, *bfp*, *traT*,

and *ompA*, across different sources, suggesting a broader adaptive advantage to thrive in diverse environments. PCA results show the genes *ompA*, *traT*, *eaeA*, and *bfp* contributed to the virulence of the isolates, and *blaTEM*, *blaCTX-M*, and *qnrs* contributed to ARGs. PCA also showed that isolates from pigs had a higher proportion of virulence genes, followed by those isolated from the environment, with those from poultry and humans having the lowest proportion, which indicates their respective potential as a reservoir for the genes. This study recommends advanced genomic analyses that could provide deeper insights into the evolutionary dynamics and transmission pathways of MDR *E. coli* across compartments, to inform more effective intervention strategies.

## Supporting information

**S1 Table. Primers for virulence genes.**
(DOCX)

**S2 Table. Multiplex PCR conditions for amplification of virulence genes.**
(DOCX)

**S3 Table. Distribution of virulence genes in MDR *E. coli* isolates.**
(DOCX)

**S4 Table. Co-occurrence of virulence genes.**
(DOCX)

**S5 Table. Association between resistance and virulence genes.**
(DOCX)

**S1 Raw images.**
(PDF)

## Acknowledgments

The authors would like to thank all members of the Department of Biochemistry & Molecular Biology and the Department of Microbiology & Immunology at MUHAS for their support during this study.

## Author Contributions

**Conceptualization:** Edwin M. James, Zuhura I. Kimera, Mecky I. Matee, Erasto V. Mbugi.

**Formal analysis:** Edwin M. James, Joely Ezekiely Efraim.

**Investigation:** Edwin M. James, Fauster X. Mgaya.

**Methodology:** Edwin M. James, Zuhura I. Kimera, Mecky I. Matee, Erasto V. Mbugi.

**Resources:** Edwin M. James.

**Supervision:** Mecky I. Matee, Erasto V. Mbugi.

**Writing – original draft:** Edwin M. James, Zuhura I. Kimera, Elieshiupendo M. Niccodem, Mecky I. Matee, Erasto V. Mbugi.

**Writing – review & editing:** Edwin M. James, Zuhura I. Kimera, Fauster X. Mgaya, Elieshiupendo M. Niccodem, Joely Ezekiely Efraim, Mecky I. Matee, Erasto V. Mbugi.

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
