## [Decision Letter · Decision Letter 0]

3 Dec 2024

PONE-D-24-37013Occurrence of virulence genes bfp, ompA, traT, eaeA, and stx1 in multidrug-resistant Escherichia coli isolates from humans, animals, and the environment: One Health perspectivePLOS ONE

Dear Dr. Niccodem,

Thank you for submitting your manuscript to PLOS ONE. After careful consideration, we feel that it has merit but does not fully meet PLOS ONE’s publication criteria as it currently stands. Therefore, we invite you to submit a revised version of the manuscript that addresses the points raised during the review process.

I have completed my evaluation of your manuscript. The reviewers recommend reconsideration of your manuscript following minor revision and modification. I invite you to resubmit your manuscript after addressing the comments below. 

We look forward to receiving your revised manuscript.

Kind regards,

Reham Mokhtar ELTarabili

Academic Editor

PLOS ONE

Journal Requirements:

2. We note that your Data Availability Statement is currently as follows: 

“All relevant data are within the manuscript and its Supporting Information files.”

5. Please upload a copy of Supporting Information Tables S1 to S5 which you refer to in your text on pages 20 to 21.

**Additional Editor Comments:**

After evaluating this manuscript well , there are some tables and titles must be merged as one table or one title and statistical analysis was missed so this manuscript must be statistically analysis .The manuscript received positive feedback from both reviewers. Reviewer #1 suggests minor revisions, including rephrasing certain lines for clarity and correcting the tense in one instance. Reviewer #2 suggests minor revision, including modifications in manner of writing. Based on the reviewers' comments, the manuscript is generally well-received but requires minor revisions for clarity and grammatical accuracy. The authors are requested to address the specific comments provided by Reviewers to improve the overall quality of the manuscript.

Reviewers' comments:

Reviewer's Responses to Questions

**Comments to the Author**

1. Is the manuscript technically sound, and do the data support the conclusions?

Reviewer #1: Partly

Reviewer #2: Yes

2. Has the statistical analysis been performed appropriately and rigorously? 

Reviewer #1: Yes

Reviewer #2: Yes

3. Have the authors made all data underlying the findings in their manuscript fully available?

Reviewer #1: Yes

Reviewer #2: Yes

4. Is the manuscript presented in an intelligible fashion and written in standard English?

Reviewer #1: Yes

Reviewer #2: Yes

5. Review Comments to the Author

Reviewer #1: Dear Editor

The manuscript "Occurrence of virulence genes bfp, ompA, traT, eaeA, 1 and stx1 in multidrug-resistant Escherichia coli isolates from humans, animals, and the environment: One Health Perspective" is quite an interesting article that comprehensively described the AMR paradigm in One Health Perspective.

Comments:

1. The names of resistance genes are not necessarily to be described in the title. It could be "Occurrence of virulence genes among multidrug-resistant Escherichia coli isolates from humans, animals, and the environment: A One Health Perspective".

2. There are lots of grammar or English mistakes in the article.

3. Novelty statement or the need of project should be more focused, need revision of Line 81. Here describe more objective of the study.

More references should be added from other parts of the world including Veterinary settings, to provide strong One Health Perspective and Global Concern.

http://dx.doi.org/10.29261/pakvetj/2023.041

https://doi.org/10.1155/2022/8224883

http://dx.doi.org/10.29261/pakvetj/2023.062

http://dx.doi.org/10.29261/pakvetj/2022.049

https://doi.org/10.1016/j.onehlt.2023.100586

4. Line 90-92, this is not required to describe the link.

5. Line 102-103: Dates are not required to be mentioned.

6. Tables are good, however, it is suggested to add at least one figure using data of Table 3.

7. Discussion section is more focused on results. Some of the critical points should be added.

Thanks and Regards

Reviewer #2: Comments to Author:

Minor points

1- In Background; page3, 49 line: You wrote virulence genes (VRs)!!! That’s

incorrect abbreviation; kindly correct it to (VGs).

2- In Background; page3, 53 line: you wrote blaOXA-4!!! That’s incorrect; kindly

correct it to blaOXA-48

3- In Background; page4, 81 line and 85 line: The rule of manuscript writing is to

avoid using (We). So you should delete (We) and use formal scientific words

(This study or The current study or The present study).

4- In Biochemical identification of the isolates; page5, 105 line: you wrote

Isolates were identified as described previously [4,26,29,31]. Briefly…..!!!

Kindly delete this sentence and directly write: The isolates were identified by

colonial morphology, Gram stain…. Do not repeat the sentences because this

makes your manuscript weak.

5- In DNA extraction for the screening of virulence genes; page6, 124 line: you

wrote whole the practical method but without writing any references!!! Kindly

mention the references that you dependent on them.

6- In PCR mixture for the detection of virulence genes; page7, 136 line: The rule

of manuscript writing is to avoid using (We). So you should delete (We) and use

formal scientific words (This study or The current study or The present study).

7- In Visualization of PCR products by electrophoresis; page7, 154 line: you

wrote whole the practical method but without writing any references again!!!

Kindly mention the references that you dependent on them.

8- In discussion; page16, 268 line, 270,283, 287, 311, 315, 321, 325, 351 and

353line: The rule of manuscript writing is to avoid using (Our) and (We). So you

should delete (Our) and (We) and use formal scientific words (This study or

The current study or The present study).

9- In discussion; page16, 274 line: you wrote the occurrence of 274 virulence

gene stx1 was low, being 4% in Karatu and 0% in this study!!! Why? You did

not suggest any reason for it!!! Try to mention accurate scientific reason.

10- In conclusion; page20, 359 line, 361 and 363line: The rule of manuscript

writing is to avoid using (We). So you should delete (We) and use formal

scientific words (This study or The current study or The present study).

11- In conclusion; page20: You should shortly mention the fact that you

explained about the PCA because this make your research stronger scientifically;

you wrote it in discussion only “PCA showed that isolates from pigs had a higher

proportion of virulence genes, followed by those isolated from the environment,

with those from poultry and humans having the lowest proportion, which indicates

their respective potential as a reservoir for the genes”.

Best regards

6. PLOS authors have the option to publish the peer review history of their article (what does this mean?). If published, this will include your full peer review and any attached files.

Reviewer #1: **Yes: **Muhammad Asif Zahoor

Reviewer #2: No

---

## [Author Response · Author response to Decision Letter 0]

31 Dec 2024

Date: 12nd December, 2024

Chief Editor,

PLOS ONE Journal

Dear Sir/Madam, 

RE: RESPONSE TO REVIEWERS COMMENTS REGARDING THE MANUSCRIPT TITLED “OCCURRENCE OF VIRULENCE GENES IN MULTIDRUG-RESISTANT ESCHERICHIA COLI ISOLATES FROM HUMANS, ANIMALS, AND THE ENVIRONMENT: ONE HEALTH PERSPECTIVE”

We sincerely appreciate the time and effort you and the reviewers have dedicated to evaluating our manuscript “Occurrence of virulence genes in multidrug-resistant Escherichia coli isolates from humans, animals, and the environment: One Health Perspective”. We are grateful for the constructive feedback provided, which has helped us improve the quality and clarity of our work. Please, find below a point-by-point response to the comments raised by the reviewers and the academic editor.

Reviewer #1 Comments:

1. Comment: The names of resistance genes are not necessarily to be described in the title. It could be "Occurrence of virulence genes among multidrug-resistant Escherichia coli isolates from humans, animals, and the environment: A One Health Perspective"

Response: The names of resistance genes have been removed in the title as suggested (Page 1, line 1)

2. Comment: There is lots of grammar or English mistakes in the article

Response: The manuscript has been checked using Quillbot, an online tool to remove grammatical errors and improve language

3. Comment: The novelty statement or the need of the project should be more focused, and need revision of Line 81. Here describe more objective of the study

Response: In addition to the originally stated objective, the statement saying; “The influence of resistance genes on the occurrence of different virulence genes was demonstrated using principal component analysis (PCA). To find out if isolates from humans, pigs, poultry, or river water had more virulence genes, PCA ellipses were employed. This study was able to assess their respective capability as gene reservoirs as a result” has been added. Page 5, Lines 86 to 92

4. Comment: More references should be added from other parts of the world including Veterinary settings, to provide a strong One Health Perspective and Global Concern.

Response: The references have been added as requested. Please refer to references number 35, 37 and 38

5. Comment: Line 90-92, this is not required to describe the link.

Response: The link has been removed as suggested (Page 5, line 96 to line 99)

6. Comment: Line 102-103: Dates are not required to be mentioned

Response: Dates have been omitted as suggested (page 5, line 102 – 106)

7. Comment: Tables are good; however, it is suggested to add at least one figure using data from Table 3

Response: Figure number 1 has been added to represent data of Table 3

8. Comment: The discussion section is more focused on results. Some of the critical points should be added.

Response: Modifications have been done throughout the discussion sections as per the suggestion

Additional Revisions: Besides addressing the reviewers' comments, we have made minor edits throughout the manuscript to improve clarity and resolve typographical errors. For your reference, these changes are detailed in the tracked-changes version of the manuscript.

We hope that our revisions meet the expectations of the reviewers and the academic editor and we look forward to your feedback.

Thank you once again for considering our work for publication in PLOS ONE.

Sincerely,

Elieshiupendo Niccodem

Lecturer

Email: eshimankamike137@gmail.com

---

## [Editor Report · Decision Letter 1]

8 Jan 2025

Occurrence of virulence genes in multidrug-resistant Escherichia coli isolates from humans, animals, and the environment: One Health Perspective

PONE-D-24-37013R1

Dear Dr. Niccodem,

We’re pleased to inform you that your manuscript has been judged scientifically suitable for publication and will be formally accepted for publication once it meets all outstanding technical requirements.

Kind regards,

Reham Mokhtar ELTarabili

Academic Editor

PLOS ONE
---

## [Editor Report · Acceptance letter]

12 Jan 2025

PONE-D-24-37013R1 

PLOS ONE

Dear Dr. Niccodem, 

I'm pleased to inform you that your manuscript has been deemed suitable for publication in PLOS ONE. Congratulations! Your manuscript is now being handed over to our production team.

Kind regards, 

on behalf of

Dr. Reham Mokhtar ELTarabili 

Academic Editor

PLOS ONE
